# Improving Geometric Validation Metrics and Ensuring Consistency with Experimental Data through TrioSA: An NMR Refinement Protocol

**DOI:** 10.3390/ijms241713337

**Published:** 2023-08-28

**Authors:** Youngbeom Cho, Hyojung Ryu, Gyutae Lim, Seungyoon Nam, Jinhyuk Lee

**Affiliations:** 1Department of Bioinformatics, KRIBB School of Bioscience, University of Science and Technology (UST), Daejeon 34141, Republic of Korea; yycho@kribb.re.kr; 2Disease Target Structure Research Center, Korea Research Institute of Bioscience and Biotechnology (KRIBB), Daejeon 34141, Republic of Korea; hjryu3094@gmail.com (H.R.); gyutae.lim@nih.gov (G.L.); 3Department of Genome Medicine and Science, AI Convergence Center for Medical Science, Gachon Institute of Genome Medicine and Science, Gachon University Gil Medical Center, Gachon University College of Medicine, Incheon 21565, Republic of Korea; 4Department of Health Sciences and Technology, Gachon Advanced Institute for Health Sciences and Technology, Gachon University, Incheon 21999, Republic of Korea

**Keywords:** protein structure refinement, validation assessment, NMR refinement protocol, torsion angle potential

## Abstract

Protein model refinement a the crucial step in improving the quality of a predicted protein model. This study presents an NMR refinement protocol called TrioSA (torsion-angle and implicit-solvation-optimized simulated annealing) that improves the accuracy of backbone/side-chain conformations and the overall structural quality of proteins. TrioSA was applied to a subset of 3752 solution NMR protein structures accompanied by experimental NMR data: distance and dihedral angle restraints. We compared the initial NMR structures with the TrioSA-refined structures and found significant improvements in structural quality. In particular, we observed a reduction in both the maximum and number of NOE (nuclear Overhauser effect) violations, indicating better agreement with experimental NMR data. TrioSA improved geometric validation metrics of NMR protein structure, including backbone accuracy and the secondary structure ratio. We evaluated the contribution of each refinement element and found that the torsional angle potential played a significant role in improving the geometric validation metrics. In addition, we investigated protein–ligand docking to determine if TrioSA can improve biological outcomes. TrioSA structures exhibited better binding prediction compared to the initial NMR structures. This study suggests that further development and research in computational refinement methods could improve biomolecular NMR structural determination.

## 1. Introduction

Protein structure determination is a fundamental task in structural biology, as it enables the understanding of protein function. Nuclear magnetic resonance spectroscopy (NMR) is one of the primary techniques widely used to determine three-dimensional (3D) protein structure. NMR spectroscopy uses the magnetic properties of atomic nuclei in a molecule to provide indirect information about its molecular structure and dynamics. This information is obtained by analyzing the resonance frequencies as prime information of the nuclei in the presence of a magnetic field. The experimental data obtained from NMR spectroscopy can be quite complex, and interpreting it requires extensive calculations and computational methods to reveal a 3D structure model [1].

NMR spectroscopy has several advantages over other methods of protein structure determination. First, it can be used to study proteins in their solution state close to their native environment, which enables NMR protein structures to be determined under near-physiological conditions [2,3], sometimes even in the cell environment [4]. Second, NMR spectroscopy can provide detailed information about the local environment specific to individual amino acid residues, which can be used to understand protein folding, stability [5], and ligand interaction [6]. Third, NMR spectroscopy can be used to study molecular dynamics beyond the static structure of proteins at various time scales, which is important for understanding their thermodynamic behaviors governing protein folding/misfolding [7] and protein–ligand binding [8].

The principal drawback of NMR in application to structural biology is the size limitation of molecules to be analyzed due to a serious decrease in sensitivity and resolution, in particular for high-molecular-weight proteins [9]. This size limitation is caused by two technical barriers. The first barrier is a spectral line broadening associated with high molecular weights. As the molecular size increases, its tumbling rate in solution decreases, leading to concomitant shortening of overall transverse relaxation time (T2), evoking NMR signals. The reduced T2 causes severe line broadening of corresponding NMR signals, which makes the signals ambiguous or even non-observable. It also reduces susceptibility to the complex pulse sequences used in NMR spectroscopy, which often require long delays for coherence transfer steps. The second barrier is the complexity of the observed NMR spectrum. As higher-molecular-weight proteins have more NMR-active nuclei and interactions, the massive number of signals with frequent signal overlaps often hinders unambiguous NMR assignments to obtain meaningful information about the molecule [10].

However, recent advances in both hardware (e.g., the availability of higher-field magnets) and experimental design promise to allow for the study of much larger proteins by pushing the current size limit of protein NMR upward to over 100 kDa [11,12]. For example, by substituting the non-exchangeable protons with deuterons (deuteration), the relaxation time of heteronuclear signals is prolonged, resulting in a narrow line width and a dramatic increase in resolution and sensitivity [13]. The availability of higher field magnets has also helped to overcome some of these limitations. Other approaches, such as solid-state NMR [14] and transverse relaxation-optimized spectroscopy (TROSY), [9] are also being used to overcome the size limitations of NMR in structural biology.

As another essential advance, computational techniques have been developed to improve calculation, particularly refinement of protein structures, based on structural restraints empirically obtained by NMR, such as chemical shift information and interatom distance restraints derived from nuclear Overhauser effect (NOE) data. The most commonly used methods for NMR protein structure calculation and refinement include distance geometry, simulated annealing, and torsion angle dynamics [15]. These methods use different approaches to improve the accuracy and precision structural determination, but they all share the same goal of finding the global minimum energy structure of the protein. The crystallography and NMR system (CNS) [16] and combined assignment and dynamics algorithm for NMR applications (CYANA) [17] software are commonly used computer software programs for determining the 3D structure of a molecule based on NMR data. These programs select the final conformers with the lowest energy and fewest violations under experimental restraints. Many NMR refinement methods combine multiple approaches to improve the accuracy of the final structure. For example, a method might use distance geometry to build a preliminary structure, then use simulated annealing to refine the structure and finally use torsion angle dynamics to further refine the structure.

In recent years, there has also been a growing interest in using deep learning to improve the accuracy of NMR protein structure refinement. AlphaFold [18] and RosettaFold [19], state-of-the-art deep-learning-based methods, have been used to model NMR structures [20] and validate or refine them with NMR chemical shift information [21]. This approach has the potential to significantly improve the accuracy of NMR protein structure refinement, especially for large or complex proteins.

Here we present TrioSA, a comprehensive NMR refinement approach that combines NOE-derived restraints, torsion angle potentials, implicit solvation models, and simulated annealing to find the lowest global energy on the empirical force field and the fewest restraint violations. TrioSA is an ensemble refinement method that was built using widely used methods to improve the accuracy of NMR protein structure determination with Chemistry at HARvard Macromolecular Mechanics (CHARMM) [22]. Distance NOE restraints and dihedral angle restraints are essential in NMR structure determination, providing experimental information on protein structure and enabling the restraints to be placed on the solution space. Adding an implicit solvation model, such as generalized born with simple switching (GBSW) [23], is crucial to account for solvent effects and improve the correctness of the final protein structures. In addition, torsion angle potentials are used to refine the ϕ-ψ, ϕ-χ1, ψ-χ1, and χ1-χ2 torsion angles of the protein structure, which can help to better capture local conformational features [24]. In the next step, SA is utilized to search for the global minimum in the empirical force field and allow the system to escape kinetic barriers, resulting in a more accurate representation of protein interactions. The combination of these components constitutes the TrioSA refinement protocol, which has the potential to re-evaluate NMR protein structure with experimental data. CHARMM has modules for NMR and solid-state NMR (SSNMR) energy functions [25]. The TrioSA refinement protocol is designed to enhance the capabilities of CHARMM in this area. It includes a preprocessing step that converts experimental NMR data into CHARMM restraint format, as well as the use of STAP potential energy. This protocol has the potential to be significantly useful in various areas of NMR research. It can improve the efficiency of NMR research. After refining the geometrical validation score, we can expect good results when applying it to various applications, making it a valuable addition to NMR structure research and encouraging further exploration and progress in the field. Currently, our TrioSA refinement protocol can utilize solution NMR experimental data. However, we have plans to expand its capabilities to incorporate solid-state NMR experimental data in the future.

## 2. Results

### 2.1. Outline of TrioSA NMR Refinement Protocol

We developed a comprehensive NMR refinement protocol called TrioSA (torsion-angle and implicit-solvation-optimized simulated annealing), which was constructed in several steps (Figure 1). First, we acquired experimental NMR data from the Biological Magnetic Resonance Data Bank (BMRB) database [26] and obtained 8741 nuclear Overhauser effect (NOE) restraints. Second, we performed preprocessing steps on the protein structures obtained from the Protein Data Bank (PDB), including consideration of disulfide bonds, removal of non-standard atoms, and conversion from CHARMM PDB format to a format more suitable for analysis [22]. These preprocessing steps were necessary to ensure that the protein structures were in a consistent format and ready for further analysis using our TrioSA refinement protocol. Detailed information about these methods is explained in the Methods section and descriptions of the supplementary figures and tables can be found in Appendix A.

The third step was to prepare the target energy terms for the NMR refinement protocol. The experimental NMR data were converted to the CHARMM restraint format. Torsion angle potentials were employed using the statistical torsion angle potentials (STAP) method to refine the ϕ-ψ, ϕ-χ1, ψ-χ1, and χ1-χ2 torsion angles of the protein structure [24]. The CHARMM molecular dynamics (MD) simulation program implemented a flat-bottom harmonic potential to constrain the dihedral angle, ensuring an accurate representation of protein interactions using the CHARMM36 all-atom force field [27]. Solvation effects were also considered using the implicit GBSW [28].

In the final stage of our study, MD simulation with SA, was used to optimize the protein structures. The target energy term in the SA procedure incorporated various experimental potentials, including NOE direct restraint, torsion angle, implicit solvation model, and stereochemical potential. The SA algorithm searches for the global minimum in the empirical force field by increasing the thermodynamic free energy of the protein system, allowing it to escape the kinetic barrier and decrease the internal energy. The system was subjected to temperature cycles from 100 K to 1000 K, then cooled to 25 K to achieve convergence. The structural quality of the refined protein structures was assessed using geometric validation scores and analysis of NOE violations in the distance and dihedral angle restraints. The NMR refinement protocol is illustrated in Figure 1, and the appropriate number of SA steps was determined by simulation tests (Appendix A).

### 2.2. Evaluating Geometrical Validation Metrics: A Comparison with RECOORD

The RECOORD database provides a comprehensive collection of optimized protein structures refined by simulated annealing calculations using the CNS and CYANA software packages [29]. It contains a curated set of structural restraints applied to over 545 proteins from the PDB analyzed for quality. RECOORD also provides explicit solvation refinement using ARIA [30], resulting in CNW and CYW sets. It serves as a valuable resource for structural biology researchers and provides a unique platform for benchmarking and comparing different NMR refinement algorithms.

We evaluated the effectiveness of the TrioSA protocol by comparing the geometric validation metrics for our 3752 refined protein structures with those for the CNS, CNW, CYA, and CYW datasets. A comparative analysis was conducted on a subset of 256 structures that overlapped between the RECOORD database, which contains a total of 545 structures, and the TrioSA refinement protocol list, which contains a total of 3752 structures. In our comparison of structural quality, we used several metrics, including normalized discrete optimized protein energy (nDOPE) [31], clash score [32], Ramachandran plots [33], and rotamer normality [34]. We based our analysis on the widely accepted nDOPE, clash score, Ramachandran plot, and rotamer normality metrics (Figure 2 and Appendix A).

The nDOPE evaluates the quality of protein models. Using a training set of high-resolution protein structures, it ranks the quality of models generated by homology modeling or ab initio prediction methods. Normalization accounts for differences in protein length and measures how well models match known structural features. Lower scores indicate higher quality, while higher scores indicate misfolding or incorrectness.

Our TrioSA-refined structure achieved the lowest nDOPE score (−0.95), outperforming CYW (−0.93), CNW (−0.93), CNS (−0.40), CYA (−0.18), and the original structures (−0.20) (Figure 2A). Explicit and implicit solvation models used in Refined, CNW, and CYW resulted in better scores than CNS and CYA without solvation models. Although CNW and CYW used explicit models, while our TrioSA refined structure used an implicit one, our score was still slightly better. In an ANOVA of the nDOPE scores of CNS, CNW, CYA, CYW, and TrioSA, a significant *p* value was obtained. However, when the analysis was conducted with only CNW, CYW, and TrioSA, no significant *p* value was observed (Appendix A).

In terms of clash score, the TrioSA-refined structure performed the best, with the lowest score of 3.09 and the lowest standard deviation, as shown in Figure 2B. CNS (21.36) showed a better clash score than CYA (59.89), while CNW (15.50) and CYW (15.80), which both include an explicit solvation model, showed similar clash score results. Both RECOORD and the TrioSA refinement protocol showed significant improvements in clash score compared to initial NMR structures (74.14).

We observed the effects of solvation models on the clash score, which measures steric clashes or repulsive forces between atoms in the protein structure; solvation models take into account the interactions of solvent molecules with the protein, which may result in lower scores, indicating a more stable protein structure with fewer steric clashes.

The Ramachandran score measures the quality of a protein structure based on the conformation of its amino acids. This score is derived from the Ramachandran plot, which shows the distribution of dihedral angles (ϕ and ψ) for each residue in the protein. The score assesses how well the ϕ and ψ angles match the preferred conformations of each amino acid as determined by experiments and simulations. A higher Ramachandran score indicates that the protein structure is more biologically relevant and likely correct, while a lower score indicates that the structure may be misfolded or incorrect. In our study, the TrioSA-refined structure (95.89) had the best Ramachandran score. As shown in Figure 2C, CNW (81.72) and CYW (81.08) achieved similar levels of improvement, while CNS (79.32) and CYA (69.08) showed either a slight improvement or deterioration compared to the initial structure (77.06). The Ramachandran plot was clearly influenced by the solvation model and the torsional angle potential (STAP) used in CNW, CYW, and the TrioSA-refined structure.

As shown in Figure 2D, the rotamer normality score evaluates the quality of protein structures based on the conformation of their side chains. This metric compares the side-chain conformations to the preferred rotamers for each amino acid type and has become important in protein docking simulations. A higher rotamer normality score indicates a more reliable side-chain conformation, as it represents a better match between the observed and expected conformations. The initial structures in this study had a low rotamer normality score of −6.21, and the scores for the RECOORD dataset of CNS (−2.35), CNW (−2.49), CYA (−7.27), and CYW (−2.78) were also below 0. However, the refined TrioSA structure had the best rotamer normality score of 2.24, highlighting the importance of this metric in protein structure analysis.

Therefore, our study provides robust evidence for the effectiveness of our TrioSA refinement protocol in improving the structural quality of proteins. Through extensive comparisons with existing methods in the RECOORD database, we demonstrated superior performance on several validation metrics, including clash score, Ramachandran score, and rotamer normality, except for nDOPE. These results suggest that our refinement protocol is a valuable tool for producing high-quality protein structures.

### 2.3. Effect of Protein Structure Refinement on NOE Restraints

Nuclear Overhauser effect (NOE) restraints provide information about the distances and relative orientations of nuclei in a molecule, which can help validate refined protein structures. In the TrioSA protocol, we fitted structures to experimental NOE restraints. By analyzing the effects of refinement on NOE restraints, we can determine whether refined structures are significantly improved over the initial NMR structures and whether they agree with experimental data. In our study, we analyzed the effects of protein structure refinement on distance NOE restraints and dihedral angle restraints. We analyzed the maximum and number of violations in various constraint categories, including all-range, intraresidual, sequential, medium-range, and long-range, for 3752 NMR protein structures. The results for the maximum violated NOEs and distance root mean square deviation (RMSD) of NMR-derived protein structures were classified into all-range, intraresidual, sequential, medium-range, and long-range distances (Appendix A). We observed significant reductions in maximum violated NOE and distance RMSD during protein structure refinement. The maximum violated NOE decreased from 1.41 (initial) to 0.71 (refined), and the distance RMSD decreased from 0.13 (Initial) to 0.07 (Refined) for all-range (Figure 3A and Appendix A). For long-range, the maximum violated NOE decreased from 1.08 (initial) to 0.50 (refined), and the distance RMSD decreased from 0.17 (Initial) to 0.08 (Refined) (Figure 3B). Our refinement protocol reduced both the number of violated NOEs and the RMSD. Figure 3C,D show that highly violated NOE decreased after refinement, while less violated NOE slightly increased for both all-range and long-range structures. In addition, our TrioSA protocol improved dihedral angle violations, with maximum violations decreasing from 28.57 (initial) to 17.12 (refined), while RMSD decreased from 0.68 (initial) to 0.52 (refined) (Figure 3E and Appendix A). Overall, our analysis demonstrates a significant improvement in the accuracy and reliability of TrioSA-refined protein structures compared to their original counterparts. We observed reduced maximum violated NOEs, distance RMSD, and dihedral angle violations. Our TrioSA refinement protocol can be applied to other NMR protein structures to improve their accuracy and reliability. The knowledge gained from this study can be used to improve our TrioSA protocol in the future, with the goal of further increasing the quality of refined NMR protein structures.

### 2.4. Contribution of Refinement Components Affecting Structural Quality Improvement

A comparative analysis was performed to evaluate the impact of TrioSA refinement components on structure quality. Five NMR refinement protocols were constructed using a combination of torsion angle potential (STAP), an implicit solvation model (GBSW), and SA, which are commonly used in protein structure refinement. The protocols were applied to 3752 TrioSA refinement datasets. The five protocols were named initial (no refinement), SA with GBSW (SA+GBSW), SA with STAP (SA+STAP), SA alone, and SA with GBSW and STAP (refined). Four validation assessments were performed on each protocol: nDOPE score, clash score, Ramachandran score, and rotamer normality (Appendix A).

The nDOPE score, which provides a normalized energy value for each non-hydrogen atom in a protein structure model, was used to evaluate the protocols, with lower scores indicating a more accurate structure. Figure 4A shows that the initial protocol had the lowest nDOPE score (−0.20), while the SA+STAP protocol had the highest score (−0.95). The SA protocol alone significantly improved the nDOPE score to −0.89 compared to the initial protocol. The scores of SA+GBSW (−0.90) and SA+STAP (−0.95) were slightly lower than that of the SA protocol. The nDOPE score of the refined protocol (−0.94) was intermediate between those of SA+GBSW and SA+STAP. The SA protocol was found to be an effective global optimization method, and STAP contributed positively to the nDOPE score, reducing it from −0.89 (SA) to −0.95 (SA+STAP). Overall, the results indicate that the SA protocol effectively improved the nDOPE score of protein structure models, with the addition of STAP further enhancing this improvement to achieve the lowest nDOPE score of all tested protocols.

The clash score, which measures the occurrence of unnatural atomic collisions, was used to evaluate the protein structure models. The results shown in Figure 3B indicate that the SA+GBSW protocol had the lowest clash score (3.39), while the initial protocol had the highest clash score (42.22). The SA protocol significantly reduced the clash score from 42.22 (initial) to 8.13. The SA+STAP protocol resulted in a slightly higher clash score of 8.64. The clash score of the refined protocol (3.86) was intermediate between SA+GBSW and SA+STAP. The solvation effect potential, GBSW, effectively reduced unnatural atomic contacts, whereas the statistical torsion angle potential function, STAP, had a slightly negative effect on the clash score. These results indicate the importance of evaluating the clash score during protein structure refinement and suggest that the SA+GBSW protocol may be the most effective in minimizing unnatural atomic collisions. The SA protocol improved significantly in reducing clash score, but the addition of STAP resulted in a slightly higher score. The refined protocol had a balanced clash score between that of the SA+GBSW and SA+STAP protocols.

The Ramachandran score was used to evaluate the backbone conformation of the protein structure models. The results presented in Figure 4C show that the Ramachandran score improved from 83.66% (initial) to 96.75% (refined), with the SA, SA+GBSW, SA+STAP, and refined protocols arranged in ascending order of score as follows: SA: 86.31%; SA+GBSW: 91.24%; SA+STAP: 95.19%; refined: 96.75%. Adding GBSW to the SA protocol improved the Ramachandran score by 4.93%, while including STAP improved it by 8.88%. The Ramachandran score was positively influenced by the ϕ-ψ torsion angle potential provided by STAP. The refined structure achieved the best Ramachandran score by integrating the benefits of SA, GBSW, and STAP, demonstrating a synergistic effect of all three components.

Rotamer normality evaluates the side-chain conformation in relation to the local minima of van der Waals and torsional potentials. This evaluation is critical to the accuracy of protein–protein/protein–chemical docking, as side-chain conformation plays an essential role in molecular interactions. The initial NMR protein structure had a rotamer normality of −4.80, indicating a negative outlier. The SA protocol slightly improved this value to −4.35, while the SA+GBSW protocol showed a moderate positive effect, with a value of −2.00. Notably, the SA+STAP protocol had a significant positive effect, with a score of −0.20, due to the inclusion of the χ1-χ2 torsion angle potential. The refined protocol showed the best rotamer normality of 2.31, highlighting the significant synergistic effect of combining all three components.

The results of the geometric validation metrics show that the combination of torsion angle, implicit solvation model, and SA significantly improved the geometric quality of the protein structures. SA, as a global optimizer, had a significant impact on the nDOPE score and the clash score. GBSW also had a positive impact on the clash score, while STAP had a significant impact on the Ramachandran score and rotamer normality. The TrioSA refined protein structures showed superior geometric quality compared to the initial structures in all validation evaluations.

### 2.5. Effect of Protein Structure Refinement on the Prediction of Binding Sites

Protein structure refinement and protein docking prediction both rely on accurate protein structures. Protein docking prediction involves prediction of the binding of one protein to another to identify potential drug targets or design new drugs. This prediction requires accurate knowledge of the three-dimensional structure of both proteins involved in the interaction. The TrioSA refinement protocol aims to optimize protein conformation to better satisfy experimental restraints and improve the overall quality of the structure.

We performed protein docking simulations using AutoDock Vina [35] and used the Matthews correlation coefficient (MCC) to evaluate the performance of 14 pairs of matching X-ray, initial, and TrioSA-refined NMR structures with ligand datasets (Figure 5 and Appendix A). A score of 1 represents perfect predictions, while a score of 0 represents completely incorrect predictions. MCC is widely used in the evaluation of protein–ligand docking predictions because it provides a fair and comprehensive assessment of a predictive model’s performance.

The MCC of the X-ray protein–ligand complex is 0.77, that of the initial NMR protein–ligand complex is 0.41, and that of the TrioSA-refined NMR protein–ligand complex is 0.49. The MCC for the TrioSA-refined NMR protein–ligand complex is 0.08 higher than for the initial NMR protein–ligand complex. Although lower than for the X-ray protein–ligand complex, our docking analysis highlights the importance of protein structure refinement in improving the accuracy of protein–ligand binding site predictions.

Therefore, our analysis provides evidence of the importance of protein structure refinement in improving the accuracy of binding predictions. The increase in MCC value from the initial to the refined structures shows that even small adjustments can have a significant impact on the accuracy of binding predictions. Our results are valuable for facilitating drug design or optimization using computational techniques that rely on docking predictions.

## 3. Discussion

Our study indicates that the TrioSA refinement protocol is a highly effective method for improving the quality and accuracy of NMR protein structures. Our analysis considered various validation metrics, including nDOPE, Ramachandran score, rotamer normality, and clash score. The TrioSA-refined structures achieved the lowest nDOPE score, the lowest clash score, the highest Ramachandran score, and the highest rotamer normality, demonstrating the potential of the refinement protocol to provide excellent geometrical validation metrics. Moreover, our study compared the geometrical validation metrics of the RECOORD datasets with those of our NMR refinement protocol. The results revealed that our refined structure outperformed the others in terms of clash score, Ramachandran score, and rotamer normality (Figure 2 and Appendix A).

We examined the NOE violation of distance and dihedral angle to confirm the changes in structural quality after NMR refinement. Experimental NOE restraints provide precision and balance between the empirical force field and experimental data [36]. Our results show that the maximum violated NOE and the distance RMSD were significantly lower in TrioSA-refined NMR protein structures compared to the initial structures in both all-range and long-range NOE. In addition, the maximum violated dihedral angle and dihedral angle RMSD were reduced in the refined structure, indicating an improvement in overall structural quality. Notably, the refined structures are more consistent with the experimental NMR data. Our results highlight the importance of long-range NOE restraints in reducing the maximum violated NOE and distance RMSD. This finding is particularly noteworthy, given that long-range distance restraints are crucial for accurate prediction of protein folding [37]. The TrioSA refinement protocol proved to be highly effective in generating accurate and stable protein structures, with important implications for the refinement of NMR-derived protein structures.

Our study demonstrated that the SA protocol significantly improved geometrical validation metrics of refined protein structures compared to their initial counterparts (Figure 2). NMR spectroscopy has emerged as a faster and more accurate method for determining protein structures. This trend is exemplified by the release of a new NMR protein structure in 2022, which exhibited remarkably high geometrical validation scores (nDOPE, −0.67; clash score, 5.98; Ramachandran score, 90.28; rotamer normality, −2.84), representing a significant improvement over initial structures (Appendix A). The SA protocol also significantly reduced the maximum violated NOE of protein structures. Currently, SA is widely utilized as the final step in most NMR protein structure determinations, and the force fields employed in SA have also undergone significant advancements in recent years. We used the CHARMM36 all-atom force field, which has been demonstrated to improve the accuracy of protein simulations, particularly in predicting the secondary structure and stability of protein complexes [38].

In this study, we investigated the impact of various refinement components on protein structure determination using five different NMR protocols. The SA+GBSW protocol, which is based on the solvation effect force of the solvent-exposed surface area for the nonpolar part with a surface tension coefficient, demonstrated the best clash score (as shown in Figure 4B). The solvation effect potential GBSW successfully reduced non-natural contact, which can cause steric clashes in the protein structure. On the other hand, the SA+STAP protocol was highly effective in improving the nDOPE score, Ramachandran score, and rotamer normality (as depicted in Figure 4A,C,D). The protein structure entropy, which is a measure of the tightness of the protein packing geometries, is derived from the backbone (ϕ-ψ) and side-chain conformation (χ1-χ2) torsion angle potential. STAP includes both the backbone and side-chain conformation potential, with the backbone conformation potential allocated to stabilize the folded protein structure and the side-chain conformation potential influencing the surface and buried residues to achieve a tight inter- and intrapacking geometry.

STAP improved Ramachandran and rotamer normality scores, essential measures of accuracy for backbone dihedral angles [39] and side-chain conformations [40,41]. The TrioSA protocol achieved the best overall validation assessment among the examined NMR refinement methods. SA, GBSW, and STAP all contributed to improving structural quality with unique features. Using these functions in the refinement process improved structural quality by allowing importance sampling with more precise restraints via conformation search. TrioSA-refined structures showed similar nDOPE, clash scores, and Ramachandran scores to X-ray protein structures but significantly better rotamer normality (Appendix A).

After applying the TrioSA refinement method, we observed an increase in the secondary structure content of the proteins. Specifically, the average alpha-helix content increased by 9.05%, the beta-sheet content increased by 8.41%, and the coil content decreased by 8.41% (Appendix A). We compared three protein structures determined by X-ray crystallography, the initial NMR, and the refined NMR structure (Figure 6A). For both pairs of structures (X-ray ID: 2PF5 vs. NMR ID: 2N40; X-ray ID: 1BRS vs NMR ID: 1AB7), we observed significant improvements in TM score and decreases in RMSD. In addition, there was an increase in alpha-helix content. For example, as shown in Figure 6A, TM score improved from 0.79 to 0.89, RMSD decreased from 2.09 Å to 1.32 Å, and alpha-helix content increased from 15.13% to 23.47%. Similarly, as shown in Figure 6B, TM score improved from 0.88 to 0.96, RMSD decreased from 1.34 Å to 0.75 Å, and alpha-helix content increased from 37.08% to 49.44%. These results demonstrate that our TrioSA refinement method effectively improves the accuracy and reliability of NMR-derived protein structures.

While geometric validation metrics are essential for evaluating protein structures, they may not always accurately capture the overall structure due to factors like protein dynamics and flexibility. In our study, we examined the relationship between geometric validation metrics and the TM score, a measure of structural similarity. Our results show a positive correlation between the two, indicating the significance of geometric validation in determining accuracy (Appendix A). However, it is worth noting that geometric validation mainly assesses local features, while the TM score considers both local and global aspects. This suggests that while geometric validation is useful for assessing local accuracy, it may not capture the complexity of protein structures in cases involving conformational changes. Further research is needed to develop methods that can better account for factors such as protein dynamics and flexibility.

The TrioSA refinement protocol has several limitations. The TrioSA refinement protocol does not provide significant methodological advances compared to standard NMR structure calculations with molecular dynamics. Moreover, the TrioSA refinement protocol is designed to provide a single, optimal structure with a high geometric validation score based on the properties of single-structure models found in X-ray structures in the PDB. Thus, this approach does not take into account the fact that NMR is sensitive to dynamics. The TrioSA refinement protocol has the advantage of improving NOE violation statistics. However, it is generally expected that using MD refinement protocols that include an experimental term in the force field leads to better NOE violation statistics. However, the improvement in NOE violations is not significant, given the large number of structures and individual restraints. Furthermore, the X-ray-based STAP method favors structures with the most typical combinations of dihedral angles. As a result, it is not surprising that refined structures exhibit better rotamer normalcy or Ramachandran scores.

However, the TrioSA refinement protocol provides an approach to re-evaluating NMR protein structures by using the latest force fields, in addition to the introduction of solvation and STAP energy potential. This allows for a more accurate and reliable assessment of the protein structure, taking into account both computational and experimental information. Additionally, we hope to contribute to future NMR research by opening our re-evaluation NMR protein structure database for comparison and analysis with other methodological software. Our database is suitable for comparison studies because it is a preprocessed list that compares all PDB-deposited NMR and BMRD-stored experimental data. Furthermore, we wrote a script that converts NMR experimental data into CHARMM constraint format, making it easy to apply through CHARMM. There is also potential for future research involving the conversion of solid-state NMR experimental data, which will allow for a more comprehensive understanding of NMR protein structures and facilitate the development of new and improved methods for their analysis.

## 4. Materials and Methods

### 4.1. Preprocessing and Selection of NMR Protein Structures

To preprocess the NMR protein structures, we converted them from the standard PDB format [42] to a format that can be read by CHARMM [22]. Specifically, we converted the NMR structures into a PDB format that was compatible with CHARMM. From the PDB, we initially downloaded a total of 12,971 NMR protein structures. These structures were then subjected to filtering based on specific parameters, resulting in the selection of 10,445 structures. The filtering criteria included a minimum requirement of more than 10 models in the ensemble and a minimum sequence length of over 20 residues. Next, we checked the availability of experimental NMR data for these structures in the BMRB [26], resulting in the selection of 7566 NMR protein structures with experimental NMR data. The NOE restraints from the XPLOR/CNS program in BMRB were then converted to a format that followed the soft asymptote potential in CHARMM. The conversion was validated using the Analyzing the QUAlity of biomolecular structures (AQUA) [43] tool, which was used to compare the results with CHARMM NOE violations. Duplicate restraints were removed using AQUA, resulting in a set of 3752 NMR protein structure entities suitable for refinement (Appendix A).

### 4.2. Energy Potential and Simulation Protocol

To better understand the NOE distance restraints, we categorized them into four groups: intraresidual, sequential, medium-range, and long-range restraints. For dihedral angle restraints, we developed a flat-bottom harmonic potential and integrated it into CHARMM for implementation. We used the empirical force field calculation formula, which defines the total energy ($E_{tot}$) of the system as the sum of the energy terms, including the stereochemistry potential ($E_{stereo}$), the implicit solvation model ($E_{solv}$), NMR experimentally derived potential ($E_{exp}$), and the torsion angle potential ($E_{STAP}$).
(1)$$\begin{array}{c} E_{tot}=E_{stereo}+E_{solv}+E_{exp}+E_{STAP} \end{array}$$

TrioSA protocols were developed using the CHARMM36 force field [27], implicit solvation model, torsion angle potential, and simulated annealing to ensure that the resulting structures closely matched the near-experimental conformation. Simulation process began by heating the system from 100 to 1000 K in 3200 steps, followed by 4000 steps of simulation at 1000 K. Subsequently, the system underwent 8000 steps of low-temperature relaxation from 1000 K to 25 K. To optimize the number of simulated annealing steps, a series of simulation tests was conducted (Appendix A).

### 4.3. Validation of NMR-Refined Protein Structures Using Comprehensive Assessment

Five types of validation scores were calculated. The first validation score was clash score, which performs an all-atom contact analysis to identify pairs of atoms that are too close together in space. The number of such pairs per 1000 atoms was calculated using MolProbity [44]. Second, Ramachandran analysis was performed to evaluate the statistical distribution of the backbone torsion angles (ϕ and ψ), which can indicate whether the protein structure has steric hindrances or is unstable. PROCHECK [45] and MolProbity were both used for this analysis. Third, the DOPE method [31], a statistical potential energy-scoring function for protein structures, was used, which was implemented in MODELLER [46]. Fourth, a dipolar distance-scaled, finite ideal-gas reference [47] was used to extract the statistical potential energy based on the distance between two atoms and the orientation angles used in dipole–dipole interactions. Finally, WHAT_CHECK [34] was used to verify protein structures and calculate the root mean square Z-score distributions of several parameters of the protein structure, including first and second packing quality, backbone conformation, and rotamer normality. By performing these comprehensive validation assessments, the geometric quality of the protein structure was analyzed from several perspectives, including steric clash, statistical potential energy, and structural quality. This approach ensured the high quality of the NMR-refined protein structures produced by the protocol and provided confidence in their accuracy for future structural analyses.

### 4.4. Evaluation of Structural Quality with NMR Experimental Restraints

The effectiveness of NMR refinement protocols was evaluated by analyzing the RMSD of the violated NOE and dihedral angle restraints. The number of violated NOEs was assessed by dividing the distance restraint range of the NOEs into 0.0, 0.5, 1.0, and 2.0 Å bins and determining the change in the number of violations. Additionally, the RMSD of the distance by which NOE upper bounds were exceeded and the dihedral angle violations were calculated. The secondary structure ratio of the protein was analyzed using the Database of Secondary Structure of Proteins (DSSP) [48] algorithm and compared to the initial and refined NMR protein structures. To evaluate how well the initial and refined NMR protein structures matched the X-ray structure, a TM score [49] analysis was conducted. A search was conducted for 583 protein structures that were available in both X-ray and NMR protein structures and had a 100% match in their amino acid sequences. These structures were then filtered based on the length of their amino acid sequences and the number of NOE restraints. Specifically, we only considered pairs of structures where the number of NOE restraints was at least five for each amino acid sequence, resulting in a total of 279 pairs of X-ray and NMR protein structures that met this criterion.

### 4.5. Docking Simulation and Evaluation of X-ray and NMR Protein–Ligand Complexes

In this study, we conducted a comprehensive analysis of 583 pairs of X-ray and NMR protein structures with identical amino acid sequences. From this dataset, we identified 143 pairs as protein–ligand complexes. To ensure a focused examination of robust complexes, we excluded ligands that were either too small or attached to the exterior of the protein. We then selected 14 protein–ligand pairs with pockets larger than 50 ${Å}^{3}$ for further study (Appendix A). Pck VMD plugin V1.0 [50] was used to select these pairs for protein–ligand docking simulations. To ensure accuracy, AutoDock Vina [35] docking simulations were performed ten times with different random seeds using a square box with a side length of 15 Å to prevent ligands from moving away from the center of the pocket residue. The protein–ligand binding energy of each clustering box was calculated using CHARMM, and the lowest-energy cluster was used to obtain protein–ligand docking residues for the X-ray, initial, and refined NMR protein structures. Finally, the accuracy of our method was evaluated by calculating the change in the MCC [51] of the initial and refined NMR protein–ligand docking residues. The MCC is a commonly used metric in bioinformatics and machine learning to estimate the quality of binary classifications, taking into account true-positive (TP) and true-negative (TN) predictions, as well as false-positive (FP) and false-negative (FN) predictions, to produce a single score ranging from −1 to 1.
(2)$$\begin{array}{c} \mathrm{MCC}=\frac{\mathrm{TP}\cdot \mathrm{TN}-\mathrm{FP}\cdot \mathrm{FN}}{\sqrt{\left(\mathrm{TP}+\mathrm{FP})\cdot (\mathrm{TP}+\mathrm{FN})\cdot (\mathrm{TN}+\mathrm{FP})\cdot (\mathrm{TN}+\mathrm{FN}\right)}} \end{array}$$

In the context of evaluating protein–ligand complexes, TP refers to correctly predicted binding events, TN refers to correctly predicted non-binding events, FP refers to incorrectly predicted binding events, and FN refers to incorrectly predicted non-binding events. We hope that this explanation helps to clarify the definition of TP, TN, FP, and FN used to calculate the MCC in the evaluation of protein–ligand complexes.

## 5. Conclusions

In this study, we aimed to develop a comprehensive NMR refinement protocol called TrioSA (torsion-angle and implicit-solvation-optimized simulated annealing). Our goal was to overcome some of the limitations of previous NMR refinement methods and provide more accurate and reliable NMR structures for various biological systems. To evaluate the effectiveness of TrioSA, we compared TrioSA-refined structures with those obtained using the RECOORD method. Our results show significant improvements in overall geometric validation metrics. Specifically, TrioSA improved geometric validation metrics such as Ramachandran plot outliers and rotamer outliers. In addition, TrioSA reduced NOE constraint violations such as maximum distance NOE violation, dihedral angle violation, and long-range NOE violation. These results indicate that TrioSA fits the experimental data better than previous methods. Our study highlights the importance of refining NMR structures using protocols such as TrioSA. By improving the accuracy and reliability of NMR-derived protein structures, TrioSA can facilitate further growth and development in NMR research. However, our study has some limitations that should be addressed in future research. For example, the TrioSA refinement protocol does not provide significant methodological advances compared to standard NMR structure calculations using molecular dynamics. Additionally, we did not conduct a comparison of TrioSA with other NMR refinement protocols. In conclusion, our study demonstrates that TrioSA is a valuable tool for NMR researchers. Its implementation can significantly improve the accuracy and efficiency of NMR structure determination. Future research should address the limitations of our study by testing TrioSA on more complex and diverse NMR datasets.

## Figures and Tables

**Figure 1 ijms-24-13337-f001:**
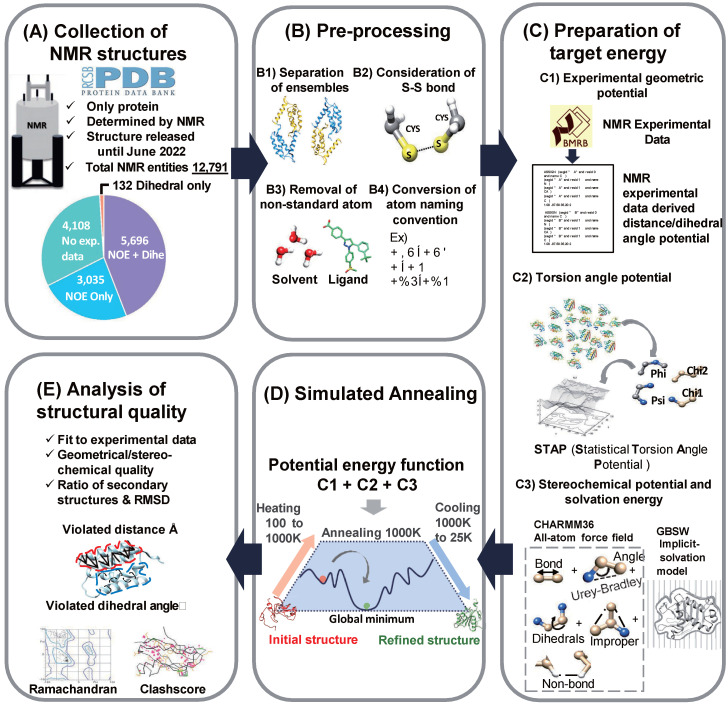
Overview of the TrioSA NMR refinement protocol. (**A**) A total of 12,971 NMR protein structures were collected from the PDB and matched to NMR experimental data from BMRB between 1986 and June 2022. (**B**) Disulfide bonds were considered, with all non-standard atoms removed before conversion to a CHARMM-restraint format. (**C**) An empirical force field was constructed using a CHARMM36 force field, NOE-directed restraints, torsion angle potential (STAP), and implicit solvation (GBSW). (**D**) Simulated annealing (SA) was performed to search for the conformation with the lowest global energy. (**E**) Geometrical validation metrics evaluated the resulting refined protein structures.

**Figure 2 ijms-24-13337-f002:**
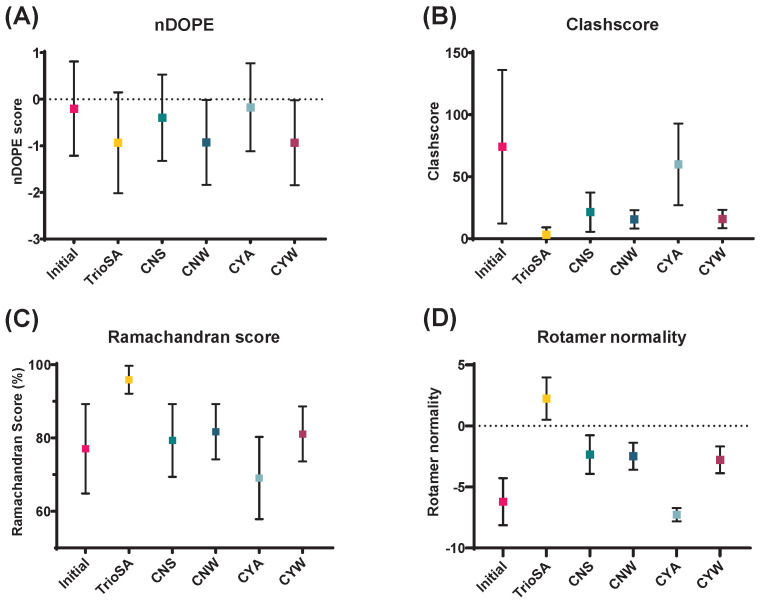
Comparative analysis of structural quality metrics before and after refinement using TrioSA and the RECOORD database. The scatter plot displays the distribution of the normalized discrete optimized protein energy (nDOPE) score (**A**), clash score (**B**), Ramachandran score (**C**), and rotamer normality (**D**) for 256 NMR structures. The error bars represent the standard deviation. The structural quality of initial and refined protein structures determined using TrioSA is compared with that obtained from popular refinement database RECOORD, including CNS, CNW, CYA, and CYW.

**Figure 3 ijms-24-13337-f003:**
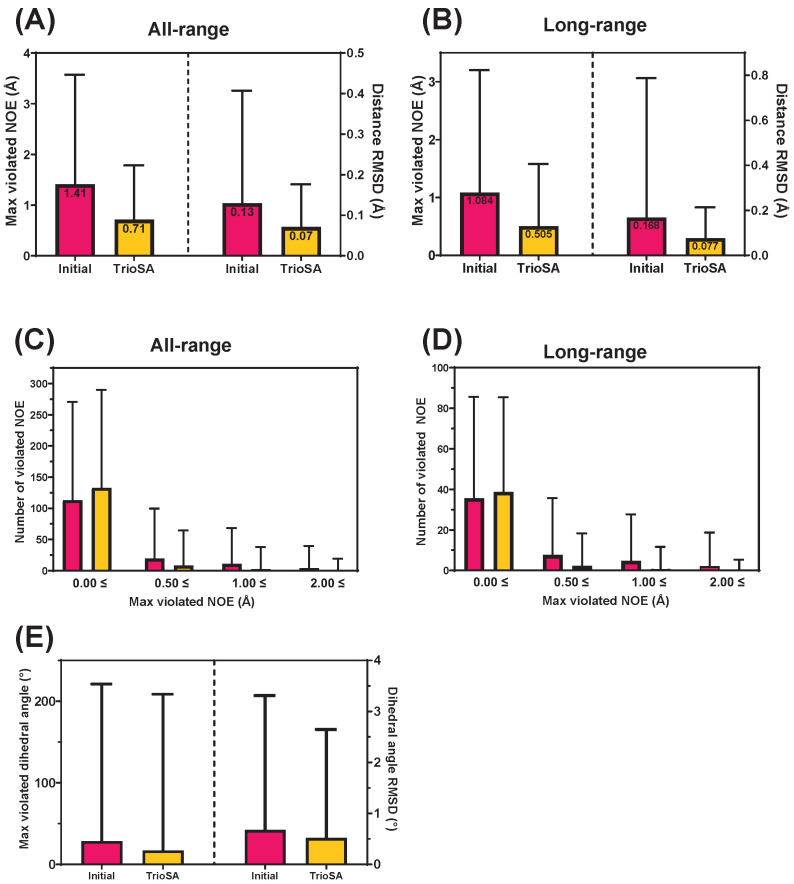
Comparison of initial and refined NMR protein structures using TrioSA. (**A**,**B**) Maximum violated NOE and distance RMSD for all-range and long-range distance NOE restraints, respectively. (**C**,**D**) The number of violated NOE restraints in the initial and refined structures, with maximum violated NOE cutoffs of 0.00 Å, 0.50 Å, 1.00 Å, and 2.00 Å for all-range and long-range distance NOE restraints. The magenta color represents the initial NMR structure, while the yellow color represents the TrioSA refined structure. (**E**) Maximum violated dihedral angle and dihedral angle RMSD histograms with standard deviation bars.

**Figure 4 ijms-24-13337-f004:**
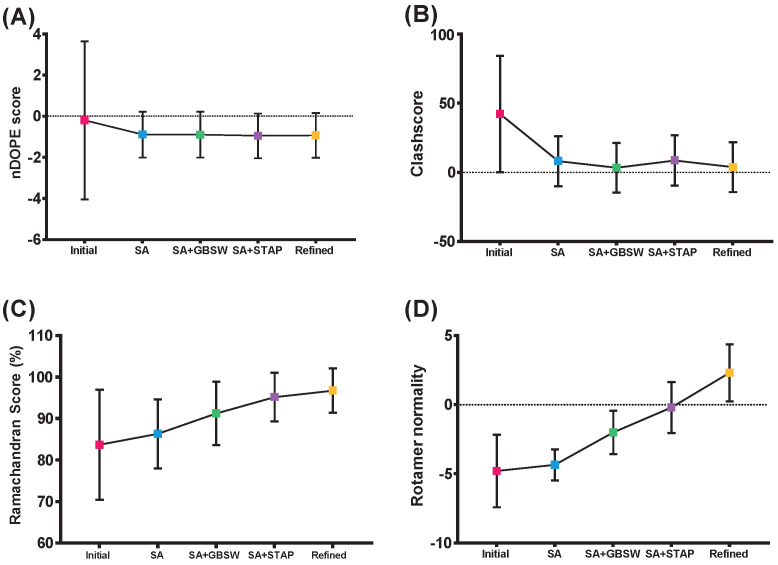
Comparison of the structural quality metrics of five NMR refinement protocols: initial, SA, SA+GBSW, SA+STAP, and refined (SA+GBSW+STAP). The histograms display the distribution of (**A**) normalized DOPE score, (**B**) clash score, (**C**) Ramachandran score (residues in the allowed region), and (**D**) rotamer normality Z score. The refinement protocols were evaluated using these metrics to determine the improvement in structural quality. The standard deviation bars indicate the variability in the data for each refinement protocol.

**Figure 5 ijms-24-13337-f005:**
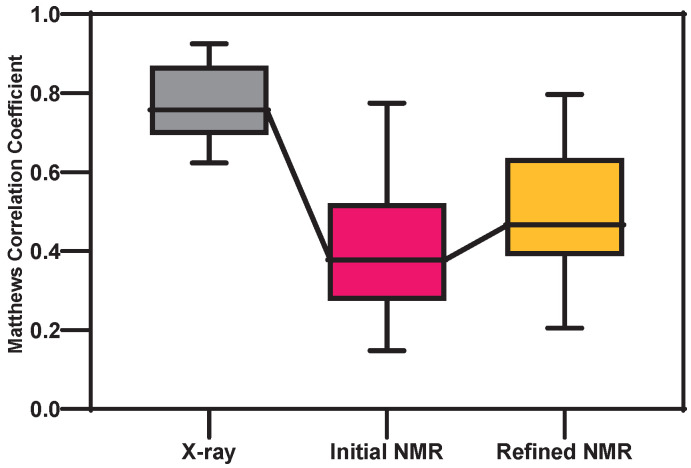
The box-and-whisker plot shows the Matthews correlation coefficient (MCC) of protein–ligand residues for the X-ray, initial, and refined NMR structures. The boxes represent the upper and lower quartiles, with the median indicated by a horizontal line within the box. The whiskers indicate the range of data, with the minimum and maximum values shown as individual points.

**Figure 6 ijms-24-13337-f006:**
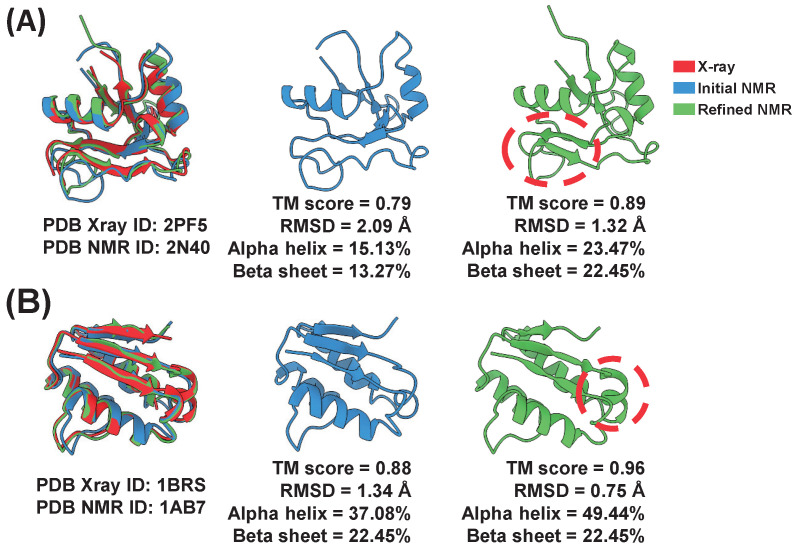
Comparison of X-ray and NMR structures in (**A**) X-ray ID 2PF5, NMR ID: 2N40; and (**B**) X-ray ID: 1BRS, NMR ID: 1AB7. The refined NMR structures (2N40 and 1AB7) displayed enhanced structural quality, as indicated by increased TM score, decreased RMSD, and increased secondary structure content compared to the original NMR structures.

## Data Availability

Data are contained within the article or Appendix A. In addition, the NMR preprocessing script, STAP, and RECOORD database related to this research have been made available at the GitHub repository: https://github.com/skyclub3/, accessed on 26 July 2023. Researchers interested in obtaining the high-capacity source files and RECOORD database can also request access by contacting yycho@kribb.re.kr via email.

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
