# Peer review of "Improving Geometric Validation Metrics and Ensuring Consistency with Experimental Data through TrioSA: An NMR Refinement Protocol"

_ijms, 2023, doi:10.3390/ijms241713337_

Round 1

Reviewer 1 Report

Structure calculation by NMR often use a typical two step protocol in which first the structure ensembles are obtained against experimental restraints typically using CNS, XPLOR or CYANA software, and later are suggested to molecular dynamics protocols that incorporate torsional, electrostatic, van der walls potentials, etc., implemented in different force fields (AMBER, CHARm, etc). The TrioSA method seems to follow a similar approach. Therefore, I found not novelty in this protocol compared with other well-stablished ones. Could the authors indicate which are the methodological advances of their methods compare the standard NMR structure calculations with molecular dynamics?

To facilitate other scientist to check the performance of the TrioSA method, the authors have to include the scripts used as part of the methods (supplementary information). This is crucial to evaluate the performance of TrioSA compared to other structural refinement methods. As a supplementary data, I think the authors should provide a list of PDB codes with the associated validation parameters shown in Figure 2, in an EXCEL like file.

The use of implicit solvent models is neither a new approach. However I miss some kind of analysis on the parametrization of the GBSW. Authors could have exploid the power of their high throughput approach to evaluate which values of GBSW lead to better results.

The comparison with the RECOORD database gives a good indication of the TrioSA method performance. Unfortunately, I couldn’t find access to this database at URLs http://www.ebi.ac.uk/msd/recoord or http://aria.pasteur.fr/aria-links/recoord-database . Thus I could not check which kind of data is available there. I being puzzle by the fact that authors use a subset of 256 structures of the RECOORD to compare. Why they did not employ the entire set of 545?. Giving that the TrioSA method is highly automated I found no reason to compare with all the RECOORD.

In summary, I don´t consider that the TrioSA method, provides any significant methodological advance to any previous method combining simulated annealing + molecular dynamics. The best value of this work is the systematic re-evaluation of the NMR structures on the PDB (with experimental data available). I think that authors should consider to focus on the potential of their protocol in automation, rather than in the improvement of the geometric quality of the structures that is always expected when using molecular dynamics force fields.

Minor points

-       Line 15: replace “ in NMR refinement could improve biomolecular structure determination” by “ in computational refinement methods could improve biomolecular NMR structure determination” ?. Because the protocol described do not involve any improvement or or addition of new NMR data.

-       Line 23: “molecules in solution.”. General NMR structure calculation references may be included here. Because it is virtually impossible to acknowledge all reviews on this topic, I would like to suggest to mention the reviews PMID: 9276248, 9794034, 28263718, or even to the K Wüthrich seminal book “NMR of Proteins and Nucleic Acids” (ISBN: 978-0-471-82893-8)

-       Also on introduction. The refining method is also potentially appliable to Solid-state NMR. I advise the authors to mention it in the introduction, and refer to relevant reviews.

-       Lines 34-41. I find this part dispensable and not well connected with the rest of the paragraph. The principal NMR limitation in structural biology is due to fast T2 relaxation associated to high molecular weights. Hence these size limitations of NMR should be also mentioned as well with the approaches to overcome them (i.e.: Solid-state NMR, TROSY or deuteration).

-       Lines 55-56. The sentence “This approach considers constraints on rotational angles to more accurately determine its 3D structure” is incorrect. Torsion angular dynamics (TAD) is just a method to sample conformation space during the simulated annealing protocol, that is alternative to cartesian dynamics. The source of restriction is not limited to torsional restraints (e.g. coming from chemical shifts or J coupling) but distance, residual dipolar couplings or others

-       can be also implemented. The TAD method do not translate these long-range constraints to torsional restrictions as it might be inferred from the sentence. Also, the TAD method does not necessary render more “accurate structure” instead it works faster than cartesian dynamics because the lower number of degrees of freedom of the system.

-       Lines 60-63. Authors should mention the RossettaFOLD approach here.

-       I found Lines 64 (from “As these methods…) to 67, dispensable

-       Line 69. “NOE-directed” should be replaced by “NOE-derived” here and elsewhere.

-       I couldn’t find access to the RECOORD database. It seems not longer accessible at  http://www.ebi.ac.uk/msd/recoord or http://aria.pasteur.fr/aria-links/recoord-database URLs.

-       Line 128. Provide references for quality metrics nDOPE and Clashscore

-       Lines 136-137. Considering the large standard deviations of the nDOPE data (Figure 2A), Is the improvement of the nDOPE of TrioSA compared to the CYW and CNW statistically relevant?. Same for other graphs in figure 2. Some kind of f-test should be included to evaluate statistical significance.

-       Figure 2B. Are error bars of yellow point missing?

-       Lines 141-142. “These results demonstrate the potential of our refinement protocol to produce highly accurate and stable refined protein structures”….accurate and stable with respect to what?

-       Lines 189-190. “protein structure refinement on distance and dihedral angle NOE constraints”. NOE constraints are only distance related, not angular. Angular constraints come from J couplings (Karplus like curves) or backbone chemical shifts.

-       Line 180 “with potential application…..drug design”. I suggest to remove for being too speculative.

-       Improvement on the NOE violations statistics (section Effect on Protein Structure … ), is generally expected when using molecular dynamics protocols that include an experimental term in the force field. The extend and number of the violations dependents on the parametrization of this force field term.

-       Line 210. I consider the sentence “Refined protein structures serve as valuable….the success of advanced methods.” Too general and therefore dispensable.

-       Line 292. “protein-ligand complex is 0.09”.  0.08?

Reviewer 2 Report

The article by Cho et al describe the evaluation of an NMR refinement protocol using simulated annealing that combines experimental constrains with energetic terms derived from CHARMM36 force field, an implicit solvation model and a statistical torsion angle potential previously developed by the same authors. They refined a large number of structures using data deposited in the BMRB database and analyzed the lowest energy structures using a number of metrics, including nDOPE, Ramachandran plots, Clashscore and violations of the experimental restraints. 

The results show that the TrioSA refinement produces structures that give comparable or better metrics than alternative protocols using explicit solvation potentials.  This is an interesting result that deserves publication but the manuscript requires substantial modifications to provide a fair image of the advantages and limitations of the method.

The introduction about what is NMR is very poorly written with possible conceptual errors. It should be revised by some NMR expert. 

The overall goal is to obtain a single “best” structure with desirable properties, mostly defined by the properties of the single structure models found in most X-ray structures in the PDB.  This approach forgets that the differential advantage of NMR is its sensitivity to dynamics, thus there is no single model that describes the reality but the goal would have to be to select the protocol that produces the best ensemble.  The lowest energy structure is one valuable piece of information, and this is what TrioSA aims for. This is fair, but the intrinsic limitations of forcing the search for structures that resemble X-ray structures should be clearly stated.

The STAP is based on X-ray structures and will favor structures that have the most common combinations of dihedral angles in these structures. It is not surprising that the refined structures have better Ramachandran scores or side chain normality (although very often little experimental information on side chain conformation and dynamics is included as experimental constraints). Also is not surprising that the nDOPE score is better as the statistic score is derived from similar type of training set.  The observation that the secondary structures are increased in the refined set is a possible problem of STAP as explicitly stated by the same authors in their 2013 paper. 

The evolution of the quality metrics with the year of publication show that the evolution of the refinement protocols (better force fields, introduction of solvation, etc.) has an important effect. This is already indicated by the authors but I could not find a deconvolution of the various aspects to ensure a fair comparison of the advantages of TrioSA as compared to alternative refinement protocols.

The description of TrioSA is clearly insufficient and mixes technical issues such as pre-processing to normalize the formats with the preparation of the energy targets, for which the form of the equations is not given explicitly nor directed to previous publications (as far as I could see). 

The improvement in NOE violations is hardly significant given the large number of structures and individual constraints. 

The evaluation of the capacity to evaluate protein-ligand complex was difficult to follow and the definition of the TP, TN, FP, FN used to calculate the MCC was not clear to me. 

Some sentences are unclear. For example

In terms of clash score, the TrioSA-refined structure showed the lowest score of 3.09 143 with the lowest standard deviation (Figure 2B). In comparison with CNS (21.36) and CNW 144 (15.50), using an explicit solvation model in the refinement by CNW resulted in better 145 performance. CYA (59.89) improved the standard deviation compared to initial NMR 146 structures (74.14) but still did not show much improvement compared to other refinement 147 structures, while CYW (15.80) and CNW had similar values.

Author Response

We have attached the RECOORD data and TrioSA refinement pre-processing script as supplementary files. Additionally, we have also included both marked-up and clean versions of the revised manuscript.
